# Modifications of Functional Human Brain Networks by Transcutaneous Auricular Vagus Nerve Stimulation: Impact of Time of Day

**DOI:** 10.3390/brainsci12050546

**Published:** 2022-04-26

**Authors:** Randi von Wrede, Timo Bröhl, Thorsten Rings, Jan Pukropski, Christoph Helmstaedter, Klaus Lehnertz

**Affiliations:** 1Department of Epileptology, University Hospital Bonn, 53127 Bonn, Germany; timo.broehl@uni-bonn.de (T.B.); thorsten.rings@uni-bonn.de (T.R.); jan.pukropski@ukbonn.de (J.P.); christoph.helmstaedter@ukbonn.de (C.H.); klaus.lehnertz@ukbonn.de (K.L.); 2Helmholtz-Institute for Radiation and Nuclear Physics, University of Bonn, 53115 Bonn, Germany; 3Interdisciplinary Center for Complex Systems, University of Bonn, 53117 Bonn, Germany

**Keywords:** epilepsy, transcutaneous auricular vagus nerve stimulation, functional brain networks, biological rhythms

## Abstract

Transcutaneous auricular vagus nerve stimulation (taVNS) is a novel non-invasive treatment option for different diseases and symptoms, such as epilepsy or depression. Its mechanism of action, however, is still not fully understood. We investigated short-term taVNS-induced changes of local and global properties of EEG-derived, evolving functional brain networks from eighteen subjects who underwent two 1 h stimulation phases (morning and afternoon) during continuous EEG-recording. In the majority of subjects, taVNS induced measurable modifications of network properties. Network alterations induced by stimulation in the afternoon were clearly more pronounced than those induced by stimulation in the morning. Alterations mostly affected the networks’ topology and stability properties. On the local network scale, no clear-cut spatial stimulation-related patterns could be discerned. Our findings indicate that the possible impact of diurnal influences on taVNS-induced network modifications would need to be considered for future research and clinical studies of this non-pharmaceutical intervention approach.

## 1. Introduction

Brain stimulation is a rapidly evolving field of research and treatment that involves different invasive and non-invasive techniques. For vagus nerve stimulation (VNS), invasive and non-invasive devices are available. Invasive VNS is an established stimulation treatment that finds application in several diseases, including depression and epilepsy. Non-invasive VNS is a more recent approach that is still under experimental and clinical investigation. Non-invasive transcutaneous vagus nerve stimulation (tVNS) can be performed as transcutaneous cervical VNS (tcVNS), percutaneous auricular VNS (paVNS), and transcutaneous auricular VNS (taVNS). The ease of use—and therefore the possibility of a rapid introduction of therapy and immediate removal of this stimulation device—explains the great interest in taVNS for research and treatment. A broad spectrum of symptoms and diseases such as CNS disorders (e.g., epilepsy [1], migraine [2], disorders of consciousness [3], and cognitive impairment [4]), as well as cardiovascular or digestive system diseases (e.g., [5,6], pain [7], insomnia [8], or COVID [9,10], are targets of taVNS. So far, the mechanism of action of taVNS is not fully understood, but widespread activity in expected vagal projection areas, including nucleus tractus solitarius, locus coeruleus, hypothalamus, thalamus, amygdala, hippocampus, as well as the prefrontal cortex and other widespread areas [11] were reported, though different study protocols and investigated subjects make interpretation difficult. There is a growing body of evidence of efficacy in several diseases [12,13]; nevertheless, there is currently no final agreement on optimal stimulation parameters. As for clinical use, it might be difficult for some patients to integrate the recommended stimulation time (e.g., for epilepsy treatment, four hours a day continuously, or as blocks of a minimum of one hour stimulation (patients’ information, tVNS technology^®^)) into their daily lives, which may lead to worse adherence to treatment advice [14,15]. Furthermore, there are currently no recommendations that state which time of the day taVNS should preferably be performed. Diurnal variation of vagal activity is a well-known phenomenon [16], and association of medical conditions linked to the vagal tone or vagal activity to certain times of the day have been demonstrated before [16,17,18,19,20]. One approach to elucidate the mechanism of action of taVNS is that it mimics an impaired or lost vagal sensory feedback to the brain [7]. Considering this ansatz and taking into account the diurnal fluctuations of the vagal activity, one can hypothesize a variation of the effect of taVNS depending on time of day.

Previous studies [15,21] demonstrated that the impact of short-term taVNS on brain dynamics can be monitored and characterised with EEG-derived evolving functional brain networks [22,23]. Stimulation-mediated modifications of various network properties indicate that short-term taVNS has a topology-modifying, robust, and stability enhancing effect. Network properties, however, may also be influenced by various biological rhythms [24], and it is not yet clear if, and to what extent, these influences impact taVNS-mediated network modifications. Addressing these issues, here, we extend our previous investigations on short-term taVNS-mediated modifications of evolving functional brain networks, and put forward the following hypotheses:modifications of the networks’ global (topology, stability, and robustness) and local characteristics (importance of network constituents) depend on the time of day the stimulation was performed; anda taVNS-related neuromodulatory effect on functional brain networks (i.e., a repeated stimulation amplifies network modifications induced by the previous stimulation) can be identified using short-term stimulations performed twice a day

## 2. Materials and Methods

### 2.1. Subjects

Subjects who were diagnosed and treated between January 2021 to July 2021 as inpatients at the Department of Epileptology, University Hospital Bonn, were screened for suitability for this study. Inclusion criteria were clinical necessity for long-term video-EEG-recording. Exclusion criteria were previous brain surgery, actual or previous neurostimulation such as invasive or non-invasive vagus nerve stimulation or deep brain stimulation, progressive disease, seizures occurring within 24 h before the start of the study or within the study, insufficient German language capability, mental disability, and incabability to follow instructions. All subjects were provided with written information and were given the opportunity to ask further questions. Eighteen subjects volunteered to participate and signed an informed consent form. The study protocol had previously been approved by the ethics committee of the Medical Faculty of the University of Bonn and was performed in accordance with the tenets of the Declaration of Helsinki.

### 2.2. Transcutaneous Auricular Vagus Nerve Stimulation and Examination Schedule

Extending previous studies [15,21], we applied transcutaneous auricular vagus nerve stimulation twice on the same day: for one hour in the mid-morning and for another hour in early afternoon while the subjects underwent continuous video-EEG-recording (morning (taVNS1): 1 h pre-stimulation phase (pre 1), 1 h taVNS phase (stim 1), and 1 h post-stimulation phase (post 1); afternoon (taVNS2): 1 h pre-stimulation phase (pre 2), 1 h taVNS phase (stim 2), and 1-h post-stimulation phase (post 2), see Figure 1).

Stimulation was carried out unilaterally in the left cymba conchae with a taVNS device (tVNS Technologies GmbH, Erlangen, Germany) with non-adjustable parameters (biphasic signal form, impulse frequency 25 Hz, impulse duration 20 s, impulse pause 30 s). The intensity of stimulation was increased until the subject noticed a non-painful “tingling”. Both stimulation phases were carried out with the same intensity. There was neither alteration of CNS medication (if taking any) nor application of activation methods (such as photo stimulation, hyperventilation, or sleep deprivation) at least 24 h before start of the study.

### 2.3. EEG Recording and Data Pre-Processing

Electroencephalograms (EEG) were recorded from 19 electrodes according to the 10–20 system (with Cz as physical reference). EEG data were sampled at 256 Hz using a 16 bit analogue-to-digital converter (Micromed, S.p.A., Mogliano Veneto, Italy) and were band-pass filtered offline between 1–45 Hz (4th order Butterworth characteristic). A notch filter (3rd order) was used to suppress contributions at the line frequency (50 Hz). All recordings were visually inspected for strong artefacts (subject movements, amplifier saturation, or stimulation artefacts), and such data were excluded from further analyses.

### 2.4. Characterising Evolving Functional Brain Networks on Global and Local Scale

Functional networks consist of vertices and edges. When analysing brain dynamics with network-theoretical approaches, vertices are usually associated with brain regions sampled by the EEG electrode contacts and edges with time-varying estimates of the strength of interactions between the vertices’ dynamics, regardless of their anatomical connections. Following previous studies [15,21], we derived evolving, fully connected, weighted, and undirected networks from a time-resolved synchronisation analysis of the abovementioned EEG-recording (sliding-window analysis on data windows of 20 s duration each), assessed important global and local characteristics of the networks, and tracked their changes over time (see [15,21] for details).

On the global network scale, we calculated the topological characteristics’ average clustering coefficient *C* and average shortest path length *L* to assess the networks’ functional segregation and integration. The former reflects independent information processes between brain regions, and the latter reflects dependent information [25]. A network’s functional segregation can be characterized by the average clustering coefficient *C*: the lower the *C*, the more segregated the network. Functional integration can be characterized by the average shortest path length *L:* the lower the *L*, the more integrated the network. In order to characterize the networks’ stability, we calculated synchronisability *S*, which assesses the networks’ propensity (or vulnerability) to be synchronised by an admissible input activation: the higher the *S*, the easier it is for the synchronised state to be perturbed. Eventually, we calculated assortativity *A*, to assess the networks’ robustness [26]. Assortativity reflects the tendency of edges to connect vertices with similar or equal properties, here weighted degree [27,28]. If edges preferentially connect vertices with similar properties, such networks are called assortative, and they tend to disintegrate into different groups more strongly than disassortative networks do. Disassortative networks are more vulnerable to perturbations and appear to be easier to synchronize than assortative networks (decreased robustness).

On the local network scale, we employed three opposing centrality concepts to characterize the role networks constituents (vertices and edges) play in the larger network [29,30,31,32,33]. A vertex or edge with a high betweenness centrality index *C^B^* is central if it connects different regions of the network as a bridge. A vertex or edge with a high eigenvector centrality index *C^E^* is central if it is connected to the vertices or edges which are central as well, reflecting the influence of the vertex or edge on the network as a whole. With strength centrality *C^S^*, the larger the sum of weights of a vertice’s adjacent edges the more central is the vertex [34,35]. For the edges, we employed the novel nearest neighbor centrality concept; a high nearest neighbor centrality index *C^N^* highlights an edge that is more central the larger its weight, and the more similar and the higher the strengths of the connected vertices [33]. Consequently, *C^N^* is largely independent of the networks’ topology as it is solely based on local vertex and edge properties. Thus, an edge with a high *C^N^* value reflects a local bottleneck possible coinciding with global bottlenecks.

### 2.5. Evaluating the Possible Influence of Biological Rhythms on Time-Dependent Network Characteristics

We estimated the power spectral density (Lomb–Scargle periodogram [36]) of time courses of local and global network characteristics (see Figure 2) to identify a possible influence of ultradian rhythms in particular, which are often defined as having periods shorter than 20 h but longer than 1 h. Given our examination schedule, we concentrated on period lengths between 30 min and 180 min. Data from subjects, for which we encountered strong contributions (spectral density > 20 [a.u.]) at these period lengths, were not taken into account for further analysis.

### 2.6. Classification of Stimulation Effects

Taking into account earlier observations [15,21], here, we define various stimulation effects acting on different timescales if network characteristics differ significantly between two phases of the examination schedule (cf. Figure 1):immediate stimulation effect: network characteristics during the pre-stimulation phase and during the stimulation phase (pre → stim) differ significantly (either in the morning or in the afternoon);enduring stimulation effect: an immediate stimulation effect can be observed and network characteristics during the pre-stimulation phase and during the post-stimulation phase (pre → post) differ significantly (either in the morning or in the afternoon);prolonged stimulation effect: an immediate stimulation effect of the morning stimulation can be observed and network characteristics during the pre-stimulation phase 1 and during the pre-stimulation phase 2 (pre 1 → pre 2) differ significantly;longer-lasting stimulation effect: an immediate stimulation effect of the morning stimulation can be observed and network characteristics during the pre-stimulation phase 1 and during the post-stimulation phase 2 (pre 1 → post 2) differ significantly. If immediate effects can be observed for both stimulations (pre 1 → stim 1 and pre 2 → stim 2), we consider the long-lasting effect to be accumulating.

A subject for whom an immediate effect of taVNS could be identified is classified as a taVNS responder (morning stimulation: taVNS1 responder; afternoon stimulation: taVNS2 responder). In addition, we refer to a network characteristic which exhibited a significant difference as annex (e.g., a subject for whom an immediate effect during the morning stimulation can be observed for the average clustering coefficient is classified as taVNS1-*C* responder).

### 2.7. Statistical Analyses

We investigated differences between network characteristics from the three phases in the morning (pre 1: pre-stimulation; stim 1: during stimulation; post 1: post-stimulation) and in the afternoon (pre 2: pre-stimulation; stim 2: during stimulation; post 2: post-stimulation) on a per-subject level using the Mann–Whitney U-test (pre 1 vs. stim 1; pre 1 vs. post 1; pre 2 vs. stim 2; pre 2 vs. post 2; pre 1 vs. pre 2; pre 1 vs. post 2; *p* < 0.05; Bonferroni correction). In order to remove possible transient effects, we neglected data from the first and last 15 min of each phase. Further downstream analyses were performed for taVNS responder only.

## 3. Results

Due to the clinical setting on the ward, recruiting participants for longer EEG-recordings without disturbing the clinically necessary work flow was challenging. From the 18 eligible subjects, three subjects had to be excluded (one due to previous seizure, one due to withdrawal of consent, one due to EEG data quality). Data from fifteen subjects (9 females; age 19–75 years, mean 40 years; duration of disease 0.1–36 years, mean 10.5 years) were included in the analyses. The same current intensities were used in both stimulation phases (range: 1.0–5.0 mA, mean 2.48, SD ± 1.2). No correlation between demographic data (age, duration of disease), as well as current intensity and immediate and enduring significant changes of network characteristics, could be observed (Pearson’s ρ *p* > 0.05).

In Figure 2a, we show exemplary time courses of the global network characteristics—average clustering coefficient *C*, average shortest path length *L*, synchronisability *S*, and assortativity *A*—of two subjects. All time courses exhibit both short-time and long-time fluctuations, albeit to varying degrees. Evaluating the possible influence of ultradian rhythms on the time-dependent network characteristics, we observed negligible contributions at period lengths between 30 and 180 min for subject 7 (see periodograms in Figure 2b). In contrast, we identified pronounced contributions at period lengths around 50, 80, and 120 min for subject 10. Similar pronounced contributions were obtained for the time courses of the global network characteristics from subjects 12 and 13. Data from these three subjects were excluded from further analysis, in order to avoid misinterpreting changes of network characteristics related to the waxing and waning of ultradian rhythms as possible taVNS-induced modifications of evolving functional brain networks.

### 3.1. Morning taVNS-Induced Immediate and Enduring Network Modifications on the Global and Local Scale

We observed in the majority of subjects an immediate stimulation effect (pre 1 → stim 1) on all global network characteristics (average clustering coefficient *C*, average shortest path length *L*, assortativity *A*, and synchronisability *S*). Depending on the investigated network characteristic, 42% to 75% of the subjects presented with significant immediate taVNS-induced modifications of their evolving functional brain networks (taVNS1-*C*: 67% (8 subjects); taVNS1-*L*: 75% (9 subjects), taVNS1-*A*: 42% (5 subjects), taVNS1-*S*: 75% (9 subjects)). Similarly, we observed an enduring stimulation effect (pre 1 → post 1) in a comparable number of subjects (*C*: 58% (7 subjects); *L*: 67% (8 subjects), A: 17% (2 subjects), *S*: 67% (8 subjects)). A small number of subjects neither responded to the morning nor to the afternoon stimulation (*C*: 2 subjects; *L*: 2 subjects and *A*: 2 subjects; note that these were not the same subjects).

Tracking the taVNS-induced modifications of networks on a single-subject level (Figure 3, left) suggested that the responders can be assigned to two subgroups, those with positive and those with negative significant modifications of their global network characteristics. Five subjects presented with an immediate increase of average clustering coefficient *C* (+8.7%, we report the change of mean values in the following) and three with an immediate decrease (−6.1%). Similarly, three subjects presented with an enduring increase of *C* (+6.2%) and another four with an enduring decrease (−4.2%). The enduring stimulation effect was, in general, less pronounced. We derived similar results for the average shortest path length *L.* Five subjects presented an immediate increase of *L* (+10.8%) and another four presented an immediate decrease (−18.1%). Four subjects presented a more pronounced enduring increase of *L* (+15.4%) and another four presented a less pronounced enduring decrease (−5.2%). For assortativity *A,* we observed an immediate decrease (−44.6%) for five subjects. Only two subjects presented enduring effects (increase in one subject, +15.6%; decrease in one subject, −31.7%). The latter figures have to be interpreted with care, given the anomalous large relative changes. In the following, we therefore refrain from an interpretation in terms of modifications of the networks’ robustness. With regard to synchronisability *S*, four subjects presented an immediate increase (+7.3%) and another five presented a more pronounced immediate decrease (−13.0%). Five subjects presented a more pronounced enduring increase of *S* (+11.2%) and another three presented a less pronounced enduring decrease (−6.2%).

On the local network scale, different vertex and edge centrality concepts highlighted different brain regions and interactions between brain regions as most central (highest centrality value), as expected. Vertex betweenness centrality highlighted left fronto-centro-temporal brain regions as most central, vertex eigenvector centrality posterior brain regions, and vertex strength centrality left temporo-parietal brain regions. Edge betweenness centrality highlighted edges as most central that connect fronto-central vertices, whereas edge eigenvector centrality as well as nearest neighbor centrality rated edges connecting left parieto-temporo-occipital vertices as most central. Note that quite often, most central edges connected vertices, one of which is also most central. Despite these distinctions, and in line with previous observations [15,21], taVNS-mediated alterations of vertex or edge centralities were presented without any clear-cut spatial pattern.

### 3.2. Afternoon taVNS-Induced Immediate and Enduring Network Modifications on the Global and Local Scale

As with the morning taVNS, we observed significant stimulation-related immediate (pre 2 → stim 2) changes to all global network characteristics (average clustering coefficient, average shortest path length, assortativity and synchronisability). Depending on the investigated network characteristic, 33% to 75% of the subjects presented significant, immediate taVNS-induced modifications of their evolving functional brain networks (taVNS2-*C*: 58% (7 subjects); taVNS2-*L*: 67% (8 subjects), taVNS2-*A*: 33% (4 subjects), taVNS2-*S*: 75% (9 subjects)). Additionally, we observed an enduring stimulation effect (pre 2 → post 2) in a comparable number of subjects for average shortest path length and synchronisability (*L*: 50% (6 subjects), *S*: 67% (8 subjects)). Significant stimulation-related enduring changes for the average clustering coefficient were observed only in two subjects, and significant changes in assortativity were observed in another subject.

We proceeded on a single subject level (Figure 3, right) and considered responders with positive and negative significant modifications of their global network characteristics. Six subjects presented with an immediate increase of average clustering coefficient *C* (+5.0%) and another subject with an immediate decrease (−5.5%). For two subjects, an enduring decrease of *C* (−1.1%) was observed. We derived similar results for average shortest path length *L*. Seven subjects presented an immediate decrease of *L* (−8.3%) and another subject an immediate increase (+20.9%). Five subjects presented a similar enduring decrease of *L* (−8.5%) and an enduring increase was observed in one subject (+25.7%, this subject also presented an immediate increase of *L*). For assortativity *A,* three subjects presented an immediate decrease of *A* (−68.2%) and one subject presented an immediate increase (+3.8%). Only one subject displayed an enduring effect (+6.3%). Again, these results have to be interpreted with care. With regard to synchronisability *S*, two subjects presented an immediate increase (+6.3%) and another seven presented a decrease (−5.0%). One subject presented a negligible enduring increase of *S* (+0.03%) and another seven presented an enduring decrease (−4.8%).

On the local network scale, we observe that most central brain regions were highlighted, most central interactions between brain regions remained unaltered, and taVNS-mediated alterations of vertex or edge centralities were again presented without any clear-cut spatial pattern.

Summarizing our findings achieved so far, both the morning and the afternoon stimulation led to immediate and enduring modifications of global network characteristics in the majority of subjects (Figure 4), but they did not specifically affect local network characteristics. These observations corroborate previous studies [15,21]. The afternoon stimulation, however, appeared to have more homogenous effects: almost all responders presented with a less segregated (increased average clustering coefficient *C*) and a more integrated (decreased average shortest path length *L*) network topology (immediate stimulation effect), and the latter (decreased *L*) even persisted into the post-stimulation phase (enduring stimulation effect). Moreover, in almost all responders taVNS increased network stability (decreased synchronisability *S*; immediate and enduring stimulation effect).

Only a subset of taVNS1 responders presented with immediate modifications of global network characteristics induced by the afternoon stimulation (50% of taVNS1-*C* responders (4 subjects), 78% of taVNS1-*L* responders (7 subjects), and 67% of taVNS1-*S* responders (6 subjects)). Comparing the direction of immediate change between morning and afternoon stimulation, only some subjects displayed immediate changes with the same direction (i.e., an increase or a decrease (*C*: 1 of 4, *L*: 2 of 7, and *S*: 3 of 6)). These numbers even decreased when considering enduring changes (*C*: 2 of 4, and *S*: 3 of 5). Thus, the hypothesized neuromodulatory effect on functional brain networks induced by short-term stimulations performed twice a day could not be identified, at least for the time scales considered here.

### 3.3. Prolonged and Longer-Lasting taVNS-Induced Modifications on the Global and Local Network Scale

About half of the subjects presented prolonged stimulation effects with regard to global network characteristics (taVNS1-*C*: 50% (6 subjects); taVNS1-*L*: 58% (7 subjects), taVNS1-*S* 50% (6 subjects). For roughly half of them, we observed network characteristics increasing, and for the other half, they decreased (Figure 4), which renders an interpretation of prolonged stimulation effects rather difficult. Breaking the data down to a single subject level, changes in the same direction were observed in all subjects for *C*, in 4 of 7 for *L*, and in 4 of 6 for *S*.

Longer-lasting stimulation effects on global network characteristics could be observed in only a small subset of subjects. Nevertheless, four of them consistently presented an increase of *C* (+6.0%). Decreased *L* was found in four subjects (*L*: −8.8%), *S* was decreased in four subjects as well (*S*: −7.3%), and increased in one subject (+15.6%). Breaking the data down to a single subject level, changes in the same direction were observed in all subjects for *C*, in 3 of 4 for *L*, and in 4 of 5 for *S*.

As expected from the abovementioned results, prolonged and longer-lasting stimulation effects on local network characteristics presented no clear-cut substructures and the most central network constituents remained unaltered.

## 4. Discussion

We employed an examination schedule consisting of two short-term transcutaneous auricular vagus nerve stimulations (one taVNS in the morning and one in the afternoon) to investigate whether taVNS-induced modifications of global and local characteristics of evolving human functional brain networks depend on time of day, and whether a neuromodulatory effect (afternoon stimulation amplifies modifications induced by the morning stimulation) can be identified. In the following, we discuss our findings obtained from twelve subjects in the light of the available research results.

### 4.1. Time-of-Day-Dependence of taVNS-Mediated Network Modifications: From Global to Local

Both the morning and the afternoon stimulation led to measurable immediate and enduring modifications of the global characteristics of the subjects’ large-scale evolving brain networks. Modifications, however, presented a clear dependence on time of day, despite our efforts to minimize the potential confounding influence of various ultradian rhythms. Whereas the afternoon-stimulation-mediated, pronounced, immediate, topology-modifying (more integrated and less segregated network), and stability-enhancing effects seen in the majority of responders corroborate previous findings [15,21] (note that in these studies, stimulations were also performed in the afternoon), pre-described enduring effects presented slightly different. This might be explained by the fact that subjects were stimulated twice, whereas in previous studies [15,21], subjects were taVNS-naïve. Interestingly, morning-stimulation-mediated modifications were rather inconsistent, despite the fact that some modifications appeared to be more strongly pronounced than the corresponding ones following the afternoon stimulation. In general, whereas pre-described immediate, enduring, topology-modifying, and stability-enhancing effects [15,21] could be observed in about 50% of responders, another 50% presented opposing modifications. Although this observation, at first glance, appears to put into perspective previous reports on taVNS-mediated modifications of global network properties, it clearly points to non-negligible influences of time of day and needs further investigation.

Contrasting the aforementioned findings, neither the morning nor the afternoon stimulation appeared to impact the most central network constituents, whose role in the larger network we rated with various opposing centrality concepts. As expected, these concepts identified different constituents as most central. In line with previous observations [15,21], our findings indicate short-term taVNS to be spatially unspecific on the local scale, thus supporting the prevalent view of a global-acting mode of action of taVNS.

Taken together, our results indicate an important influence of time of day on taVNS-mediated modifications of various properties of evolving functional brain networks. This emphasizes not only the need to report the time of day of stimulation as recommended previously [37], but also the necessity to consider diurnal variations for the interpretation of research findings and clinical trials, as well as for the formulation of application recommendations. It is, however, conceivable that there are additional confounding factors that affect the results of taVNS stimulation, such as habituation effects.

Thus, at first glance, counter-intuitive observations of unaffected local properties but strongly-affected global ones, which, additionally appear to be dependent on the time of day, can be reconciled well with an extension of the previously suggested model of a stimulation-induced stretching and compression of the functional brain network (see [21] for a detailed description of the model). With this model, taVNS-mediated modifications of the larger networks are characterized by taking into account the changes of the network’s path structure (average shortest path length), its tendency to form tightly knit groups of vertices (average clustering coefficient), and the importance of hierarchies of vertices and edges. Here, we observed the dynamics of this model for the vast majority of responders with the afternoon stimulation, and for about half of the responders with the morning stimulation. Since the other half of the responders in the morning stimulation resulted in a reversed pattern, this might point to a sensitive dependence on time of day. With the extension of the model proposed here (Figure 5), we propose that this dependence can be characterized by some rhythmic activities that interfere with the stimulation sequence. These activities may predominantly represent biological rhythms with different period lengths (ultradian and/or circadian rhythms), diurnal fluctuations of the vagal activity, as well as superpositions thereof.

### 4.2. Prolonged and Longer-Lasting taVNS-Mediated Network Modifications

Our approach allowed us to assess taVNS-mediated network modifications on shorter (immediate and enduring effects) and longer time scales (prolonged and longer-lasting effects). For the latter, we solely observed modifications of global network properties, whereas local ones remained largely unaltered. Prolonged effects resembled, to a large extent, the immediate effects seen for the morning stimulation, and in general, few responders presented inconsistent modifications of their functional brain networks. For an even smaller subset of responders, we observed longer-lasting effects, and these subjects presented network modifications similar to the ones observed as immediate effects following the afternoon stimulation. Nevertheless, given that the majority of subjects did not present longer-lasting effects, potential factors affecting short-term taVNS-mediated network modifications on longer time scales remain to be identified. In this regard, a comparison with previous research findings is only of limited value. Although immediate effects of short-term taVNS on brain dynamics have been reported repeatedly in healthy and different medical conditions, so far information on potential effects acting on time scales that range from hours to years can only be derived from clinical studies in different diseases [12,13,38]. These studies, however, are largely based on repeated daily stimulations over long periods of time, and there might be other potential influencing factors, such as pharmaceutical treatment, that would need to be taken into account when interpreting stimulation-related modifications of brain dynamics. In addition, in these studies, taVNS-mediated effects are assessed only indirectly via the clinical outcome (e.g., seizure frequency, headache scores, depression scores). During the time after the stimulation and over subsequent days, knowledge about taVNS-mediated effects on the human brain is sparse. We expect, however, that ultradian rhythms with period lengths longer than the ones considered here, as well as circadian or even infradian rhythms, would need to be considered as potential confounders when investigating long-lasting taVNS-mediated modifications of evolving functional brain networks.

### 4.3. Can a Neuromodulatory Effect of Short-Term taVNS Be Identified?

In line with previous studies [15,21], we observed significant immediate short-term modifications of topology- and stability-related network properties in up to three quarters of investigated subjects. A subset of those subjects presented as responders to both stimulations, but the percentage of responders to both stimulations was essentially the same. With our examination schedule (one stimulation in the morning and one in the afternoon, with a heuristically chosen gap between them) and classification of stimulation effects, we could not identify a potential neuromodulatory effect (the second stimulation amplifies network modifications induced by the first stimulation); however, an influence cannot be excluded by our design, since even non-significant changes resulting from the first stimulation might serve as amplifier for the second one. There is experimental (cf. [39]) and clinical evidence (for example, from studies in epilepsy, cf. [13]) for vagal nerve stimulation (VNS) to have a neuromodulatory effect that increases over time. As the vagal nerve itself has a transmitting and not a processing function [39], amplifying effects due to repeated stimulations are assumed to act on the molecular, neurotransmitter, and synaptic level. Though the exact mechanism of action is not fully understood, a growing body of evidence for VNS-induced alterations of different transmitter pathways is available, and to some extent, can be extended to taVNS [7]. It is supposed that a certain—not yet fully determined—threshold must be exceeded to achieve neuromodulatory effects of taVNS. Future studies would need to identify the necessary amount, duration, and period of time of stimulations along with their temporal arrangement that are required to surpass the hypothesized threshold.

To conclude, our findings point to an important influence of time of day on taVNS-mediated modifications of various properties of evolving functional brain networks. Future studies should address the influence with a double-blind approach (stimulation versus sham stimulation) and interpersonal work-up (morning stimulation versus afternoon stimulation group) to further corroborate our findings. Future studies should also investigate a possible relationship between taVNS-induced modifications of functional brain networks and clinical efficacy, which is needed to translate experimental results into clinical decision making. Importantly though, future clinical trials should take into account the potential diurnal influence. Considering the time of day when vagal stimulating is supposed to be most effective, and might not only increase the clinical outcome, but might also allow the stimulation time to be reduced, which could increase adherence. Investigating taVNS-mediated modifications of evolving human functional brain networks on longer time scales is necessary to better understand the mechanism of action in taVNS, as well as to establish meaningful protocols for research and treatment trials.

## Figures and Tables

**Figure 1 brainsci-12-00546-f001:**
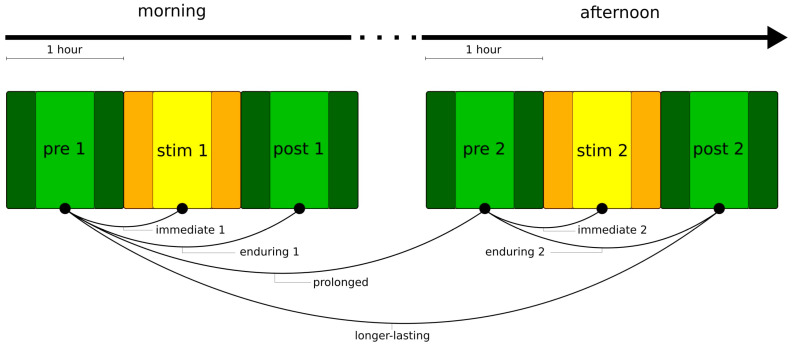
Examination schedule to probe for short and longer-lasting taVNS-induced changes in evolving functional brain networks. The schedule consisted of two 1 h stimulation phases (morning: stim 1 and afternoon: stim 2; yellow-shaded blocks) each phase was preceded and followed by a pre- and post-stimulation phase, each lasting one hour (green-shaded blocks). During each of the 3 h blocks, subjects continued laid-back activities (no other activation methods applied, no eating) and they continued performing daily activities during the break (no other activation methods applied). The whole examination schedule (including the break) was embedded into a continuous video-EEG-recording. In our analyses, we neglected data from the first and last 15 min of each phase (darker colours) in order to remove possible transient effects (e.g., due to movements or expectation effects).

**Figure 2 brainsci-12-00546-f002:**
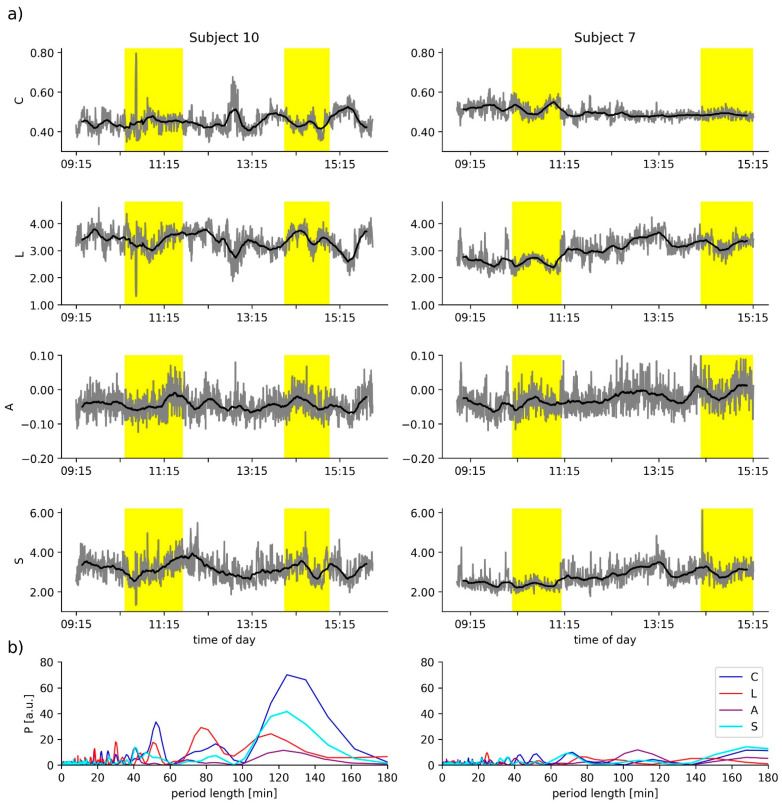
(**a**) Exemplary time courses (grey lines) of average clustering coefficient *C*, average shortest path length *L*, assortativity *A*, and synchronisability *S* of two subjects. Smoothed time courses (moving average over 17 min) are shown as black lines. Yellow-shaded areas mark the two 1 h stimulation phases. (**b**) Power spectral density estimates *P* of the respective time courses.

**Figure 3 brainsci-12-00546-f003:**
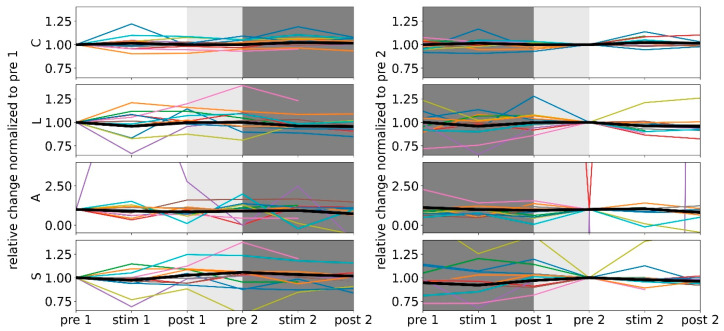
Relative changes of global network characteristics (average clustering coefficient *C*, average shortest path length *L*, assortativity *A*, and synchronisability *S)* of all responders (colour-coded). Group medians are shown in black, and all lines are for eye-guidance only. Data normalized to the respective values from the morning pre-stimulation phase (**left**; white-shaded area) and from the afternoon pre-stimulation phase (**right**; white-shaded area). Light-grey shaded area marks the break between the two 3 h examination phases. The different scaling of the *y*-axis for assortativity is due to the fact that this characteristic only rarely deviated from −0.02 which resulted in anomalous large relative deviations. We note that we obtained for the vast majority of investigated networks, indications for a random topology (their assortativity values were confined to the range of *A* values derived from 1000 random networks with the same number of vertices and edge densities as the evolving functional brain networks).

**Figure 4 brainsci-12-00546-f004:**
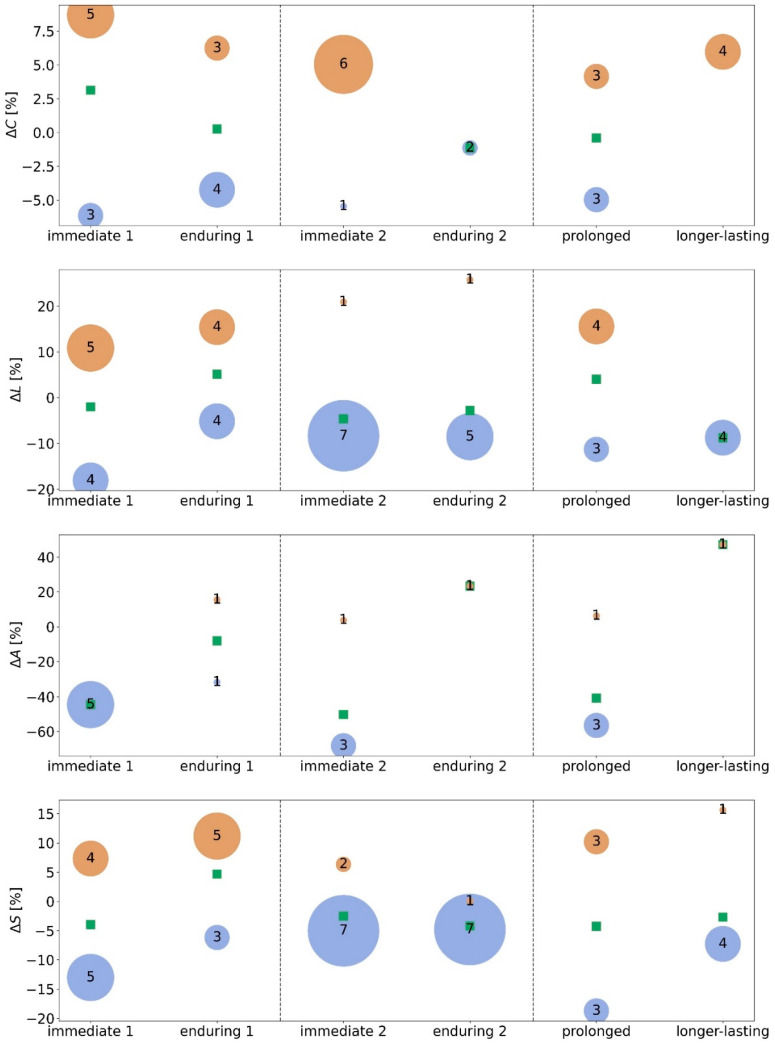
Bubble chart of taVNS-induced immediate, enduring, prolonged, and longer-lasting relative changes Δ of global network characteristics of all responders (average clustering coefficient *C*, average shortest path length *L*, assortativity *A*, and synchronisability *S*; Δ = (*M*_l_ − *M*_k_)/*M*_k_, where *M*_k_ and *M*_l_ denote placeholders for the temporal average of the respective characteristics from phase k and phase l; cf. Figure 1). Responders are assigned to subgroups according to their direction of change (positive/negative changes are shown in orange/blue). A disk is centred at a subgroup’s mean relative change, and the diameter of a disk encodes the number of subjects per subgroup. Mean relative changes from both groups are shown as green squares.

**Figure 5 brainsci-12-00546-f005:**
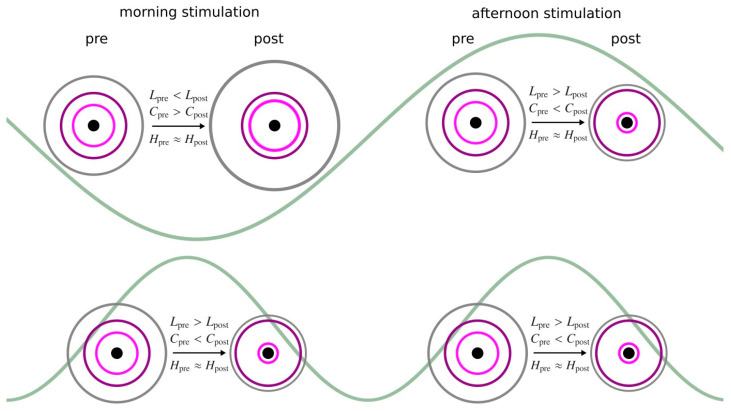
Schematic of spatial networks prior to and after taVNS stimulation in the morning (**left**) and in the afternoon (**right**), with different rhythmic activities (longer period lengths: top, shorter period lengths: bottom; indicated by greenish waves in the background) that interfere with the stimulation-mediated network modification. The network is separated into different areas based on network properties: (average clustering coefficient *C* and average shortest path length *L*): boundary (gray outermost ring), periphery (purple and pink rings in the middle), and core (black innermost ring/circle). The larger the radius of a ring, the higher the *L* and the smaller the *C* of vertices in this area. The closer two rings, the more clustered the vertices are in the two areas relative to each other. Local importance hierarchies *H* (assessed with different centrality concepts) of vertices and edges are not affected by the stimulation.

## Data Availability

The data presented in this study are available on request from the corresponding author. The data are not publicly available as they contain information that could comprise the privacy of research participants.

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
