# Peer review of "Modifications of Functional Human Brain Networks by Transcutaneous Auricular Vagus Nerve Stimulation: Impact of Time of Day"

_brainsci, 2022, doi:10.3390/brainsci12050546_

Round 1

Reviewer 1 Report

Modifications of functional human brain networks by transcutaneous auricular vagus nerve stimulation: impact of time of day does an excellent job detailing the authors findings regarding diurnal variability of VNS response, and highlights the importance of time-of-day to the neurophysiologic parameters studied. Despite the need for further investigation into the mechanism of action for VNS in general (not just taVNA), which the correctly authors point out, the clinical significance of the findings of the authors is debatable, and overall unclear. Correlating the parametric findings to clinical outcomes (such as seizure reduction response, or even lifestyle metrics, Engel score, etc.) would help with this shortcoming of this paper to make it more relevant to epileptologists, neurologists, neurophysiologists, and neurosurgeons. Overall, this paper adds to a small body of literature investigating the network-responsiveness of VNS stimulation, and, while lacking in direct clinical relevance, its contribution to the study of VNS is important and thorough enough that it should be published.

Author Response

We thank the reviewer for the time s/he spent evaluating our manuscript.

Despite the need for further investigation into the mechanism of action for VNS in general (not just taVNA), which the correctly authors point out, the clinical significance of the findings of the authors is debatable, and overall unclear. Correlating the parametric findings to clinical outcomes (such as seizure reduction response, or even lifestyle metrics, Engel score, etc.) would help with this shortcoming of this paper to make it more relevant to epileptologists, neurologists, neurophysiologists, and neurosurgeons.

We thank the reviewer for the comment. We would like to point out that our investigations are still on a basic research level, and we are not yet ready to make statements about the clinical significance of our findings. We rewrote parts of the manuscript to emphasize the – by now - experimental approach and the need for future investigations on taVNS-induced changes in functional brain networks and clinical efficacy.

Reviewer 2 Report

 The authors have designed an experiment to examine those properties of networks extracted from EEG recordings which are influenced by the application of taVNS to epilepsy patients, and they showed that the influence on some of these characteristics by the stimulation was different when stimulation was provided at different times of day. Each patient was given stimulation both in the morning and again in the afternoon on the same day. A range of network characteristics were calculated to examine the effects on both local and global network properties, and effects posited to be caused by the time of day of the stimulation were found in the global metrics but not the local ones. 

In it's current form the MS does not clearly identify and discuss possible confounds for their claim that time of day is the cause for the differences in response to stimulation that they report. With this experimental design, I do not find convincing the authors argument that the time of day is the major cause of differences found in response to the two stimulation sessions. Relating to the second of the two hypotheses, the authors state in the discussion that "we could not identify a potential neuromodulatory effect (the second stimulation amplifies network modifications induced by the first stimulation)". From this the inference appears to be that changes in the response to stimulation in the afternoon must be due to inherent differences in the "vagal system" that are due soley to the time of day of the stimulation. Although a comparison is done between the two baselines ('pre1' and 'pre2'), an absence of a statistically significant difference between them does not preclude an influence of the earlier session on the observed differences between pre2 and stim2 [Ref 1]. I find this to be the major problem with this study design, which could have been addressed by having two groups of patients stimulated at different times of day. Discussion/more clarity around this issue should be provided. (An even more mundane factor could perhaps be changes in skin conduction at the stimulation site due to the earlier session, leading to different stimulation levels between sessions).

Specific comments :

In the statement of hypotheses (lines 69-73), it is not clear what exactly is meant by neuromodulatory effect. This is clarified in the discussion (line 365,459), but should be mentioned at the original hypothesis statement. 

Line 102: What 'transient effects' might be controlled for by removing the last 15 minutes of each session? 

Line 139: "Assortativity reflects the tendency of edges to connect vertices with similar or equal properties." - which 'properties'? weighted degree? C,L,S ? Please explicitly state this. If C,L,S what is the metric to measure similarity for a pair of vertices?  

Lines 140-144 - it is not clear to me from these two sentences whether increased assortativity is related to increased or decreased 'robustness'. Robustness to what? A little more clarity here would be good. Please also provide a citation which relates assortativity to network robustness. 

Line 239: Were the 'non-responders' the same 2 subjects for C,L and A? 

I find the model description (last paragraph p5) and Figure 5 difficult to understand. The authors state that the "...network is separated into different areas" (line 419). Are these topological regions? Physical areas? It is not clear to me what this means, or what the circle sizes represent. More clarity here would be helpful. 

Ref: 

[1] Nieuwenhuis, S., Forstmann, B. & Wagenmakers, EJ. Erroneous analyses of interactions in neuroscience: a problem of significance. Nat Neurosci 14, 1105–1107 (2011). https://doi.org/10.1038/nn.2886 

Author Response

We thank the reviewer for the time s/he spent evaluating our manuscript.

In it's current form the MS does not clearly identify and discuss possible confounds for their claim that time of day is the cause for the differences in response to stimulation that they report. With this experimental design, I do not find convincing the authors argument that the time of day is the major cause of differences found in response to the two stimulation sessions. Relating to the second of the two hypotheses, the authors state in the discussion that "we could not identify a potential neuromodulatory effect (the second stimulation amplifies network modifications induced by the first stimulation)". From this the inference appears to be that changes in the response to stimulation in the afternoon must be due to inherent differences in the "vagal system" that are due soley to the time of day of the stimulation. Although a comparison is done between the two baselines ('pre1' and 'pre2'), an absence of a statistically significant difference between them does not preclude an influence of the earlier session on the observed differences between pre2 and stim2 [Ref 1]. I find this to be the major problem with this study design, which could have been addressed by having two groups of patients stimulated at different times of day. Discussion/more clarity around this issue should be provided. (An even more mundane factor could perhaps be changes in skin conduction at the stimulation site due to the earlier session, leading to different stimulation levels between sessions).

We thank the reviewer for diligently considering our hypotheses and statistical approach. We agree that some of our previous statements were too strong. We rewrote the respective parts of the manuscript and weakened some of our claims. By now, we observed differences between morning and afternoon stimulation, and while one explanation might be the influence of the time of day, other confounders are of course conceivable. We hope we could clarify this in the revised manuscript. We also rewrote the paragraph on neuromodulatory effects and hope, that we now clarified that the missing identification of potential effects is not meant as an exclusion of these effects.

In the statement of hypotheses (lines 69-73), it is not clear what exactly is meant by neuromodulatory effect. This is clarified in the discussion (line 365,459), but should be mentioned at the original hypothesis statement.

We thank the reviewer for pointing out the missing information, which we now added to the Introduction section.

Line 102: What 'transient effects' might be controlled for by removing the last 15 minutes of each session?

As the participants were informed of the study protocol, an arousal effect due to expectation at the end of the announced stimulation cannot be ruled out and could confound stimulation effects. We now clarify this in the revised manuscript.

Line 139: "Assortativity reflects the tendency of edges to connect vertices with similar or equal properties." - which 'properties'? weighted degree? C,L,S ? Please explicitly state this. If C,L,S what is the metric to measure similarity for a pair of vertices? 

The property in question was indeed the weighted degree. We now clarify this in the revised manuscript.

Lines 140-144 - it is not clear to me from these two sentences whether increased assortativity is related to increased or decreased 'robustness'. Robustness to what? A little more clarity here would be good. Please also provide a citation which relates assortativity to network robustness.

We thank the reviewer for pointing out this missing information. We rewrote the respective part of the manuscript and added the requested references. We refrained, however,  from a more detailed explanation as assortativity is already described in detail in the cited studies. Likewise, the other network characteristics would have also to be described in more detail in order to achieve the balance. This would lead to an unnecessary expansion of our manuscript.

Line 239: Were the 'non-responders' the same 2 subjects for C,L and A?

No, the non-responder were not the same subjects. We added this information to the revised manuscript.

I find the model description (last paragraph p5) and Figure 5 difficult to understand. The authors state that the "...network is separated into different areas" (line 419). Are these topological regions? Physical areas? It is not clear to me what this means, or what the circle sizes represent. More clarity here would be helpful.

The model of a stimulation-induced stretching and compression was introduced just recently (Rings et al., Frontiers in Physiology 12,700261, 2021) and we have therefore taken the liberty of referring to the details of this work. A simplified summary can be found in the figure caption.

Reviewer 3 Report

I thank the authors for their remarkable efforts in conducting this interesting study. Though there are some methodological limitations, I think this paper is of major interest both for basic science and clinical applications.

Despite efforts to minimize the role of the biological clock in the interpretation of the observed differences, I have one major comment that unfortunately cannot be resolved at this stage: the absence of sham stimulations.

Minor comments

  • My second major comment concerns the specific role of time of day versus the potentiation effect of Stimulus 1 on Stimulus 2. In the opinion of the authors, it appears that afternoon stimuli affect cortical connectivity more because of the time of day. However, according to the protocol, these early afternoon stimulations are conducted after a mid-morning stimulation without the authors specifying whether the two stimulations are done the same day. The authors took care to consider a minimum time of 24 hours between the last seizure and the first stimulation. Such a minimum time interval should have also been observed between the two stimulation sessions. In the worst-case scenario, since stimulation 1 can significantly increase the cortical response to stimulation 2, stimulation 1 should have randomly been conducted in the afternoon for some patients and in the morning for others. Although it is too late to correct this methodological bias, I think the authors should better discuss this point. The discussion of this issue in section 4.3 is insufficient in my opinion.
  • Section 2.3: please specify the software used for the EEG.

Author Response

We thank the reviewer for the time s/he spent evaluating our manuscript.

Despite efforts to minimize the role of the biological clock in the interpretation of the observed differences, I have one major comment that unfortunately cannot be resolved at this stage: the absence of sham stimulations.

We are aware of this limitation and added this point in the Discussion section.

My second major comment concerns the specific role of time of day versus the potentiation effect of Stimulus 1 on Stimulus 2. In the opinion of the authors, it appears that afternoon stimuli affect cortical connectivity more because of the time of day. However, according to the protocol, these early afternoon stimulations are conducted after a mid-morning stimulation without the authors specifying whether the two stimulations are done the same day. The authors took care to consider a minimum time of 24 hours between the last seizure and the first stimulation. Such a minimum time interval should have also been observed between the two stimulation sessions. In the worst-case scenario, since stimulation 1 can significantly increase the cortical response to stimulation 2, stimulation 1 should have randomly been conducted in the afternoon for some patients and in the morning for others. Although it is too late to correct this methodological bias, I think the authors should better discuss this point. The discussion of this issue in section 4.3 is insufficient in my opinion.

We thank the reviewer for pointing to this oversight on our part. The two stimulations were indeed applied on the same day. We clarified this in the revised manuscript and furthermore rewrote the Discussion section to consider possible confounders and biases.

Section 2.3: please specify the software used for the EEG.

We added this information to the Methods section.

Round 2

Reviewer 2 Report

The authors have addressed the points that I raised satisfactorily.